# Nineteen-year prognosis in Japanese patients with biopsy-proven nonalcoholic fatty liver disease: Lean versus overweight patients

Shunji Hirose[1]*, Koshi Matsumoto[2], Masayuki Tatemichi[3], Kota Tsuruya[1], Kazuya Anzai[4], Yoshitaka Arase[5], Koichi Shiraishi[6], Michiko Suzuki[1], Satsuki Ieda[1], Tatehiro Kagawa[1]

1 Department of Gastroenterology and Hepatology, Tokai University School of Medicine, Isehara, Japan,
2 Department of Pathology, Ebina General Hospital, Ebina, Japan, 3 Department of Preventive Medicine,
Tokai University School of Medicine, Isehara, Japan, 4 Department of Gastroenterology and Hepatology,
Tokai University Hachioji Hospital, Hachioji, Japan, 5 Department of Gastroenterology and Hepatology, Tokai
University Oiso Hospital, Nakagun, Japan, 6 Department of Gastroenterology and Hepatology, Tokai
University Tokyo Hospital, Tokyo, Japan

* s-hirose@is.icc.u-tokai.ac.jp

**Data Availability Statement:** All relevant data are within the manuscript and its Supporting Information files.

## Abstract

### Background

Many studies have investigated the prognosis of nonalcoholic fatty liver disease (NAFLD); however, most studies had a relatively short follow-up. To elucidate the long-term outcome of NAFLD, we conducted a retrospective cohort study of patients with biopsy-proven NAFLD.

### Methods

We re-evaluated 6080 patients who underwent liver biopsy from 1975 to 2012 and identified NAFLD patients without other etiologies. With follow-up these patients, we evaluated the outcome-associated factors.

### Results

A total of 223 patients were enrolled, 167 (74.9%) was non-alcoholic steatohepatitis (NASH). The median follow-up was 19.5 (0.5–41.0) years and 4248.3 person-years. The risk of type 2 diabetes mellitus (T2DM) and hypertension was 11.7 (95% confidence interval [CI] 8.70–15.6) and 7.99 (95% CI 6.09–10.5) times higher, respectively, in NAFLD patients than in the general population. Twenty-three patients died, 22 of whom had NASH. Major causes of death were extrahepatic malignancy and cardiovascular disease (21.7%) followed by liver-related mortality (13.0%). All-cause mortality was significantly higher in NASH patients than in nonalcoholic fatty liver patients (P = 0.041). In multivariate analysis, older age (hazard ratio [HR] 1.09 [95% CI 1.05–1.14], P<0.001) and T2DM (HR 2.87 [95% CI 1.12–7.04], P = 0.021) were significantly associated with all-cause mortality. The factors significantly associated with liver-related events were older age, T2DM, milder hepatic

**Funding:** T.K. was supported, in part, by a Grants-in-Aid for Scientific Research (C) https://www.jsps.go.jp/english/e-grants/ (18K11002) The funders had no role in study design, data collection and analysis, decision to publish, or preparation of the manuscript. The other authors declare that they do not have anything to disclose with respect to the funding, sources of support or conflict of interest (whether external or internal to our organization) received during this study. There was no additional external funding received for this study.

**Competing interests:** The authors have declared that no ompeting interests exist.

**Abbreviations:** AIH, autoimmune hepatitis; BMI, body mass index; CI, confidence interval; FIB4, fibrosis-4; HBV, hepatitis B virus; HCV, hepatitis C virus; HCC, hepatocellular carcinoma; HR, hazard ratio; NAFL, nonalcoholic fatty liver; NAFLD, nonalcoholic fatty liver disease; NASH, nonalcoholic steatohepatitis; PNPLA3, patatin-like phospholipase domain containing 3; PBC, primary biliary cholangitis; PSC, primary sclerosing cholangitis; PYF, person-years of follow-up; SIR, standardized incidence ratio; T2DM, type 2 diabetes mellitus; WHO, World Health Organization.

steatosis, and advanced liver fibrosis. Body mass index wasn't associated with either mortality or liver-related events.

## Conclusions

T2DM was highly prevalent in NAFLD patients and was significantly associated with both all-cause mortality and liver-related events. The lean patients' prognosis wasn't necessarily better than that of overweight patients.

## Introduction

Nonalcoholic fatty liver disease (NAFLD) is a hepatic phenotype of metabolic syndrome [1–5]. NAFLD is classified into nonalcoholic fatty liver (NAFL) and nonalcoholic steatohepatitis (NASH). NAFL does not involve hepatic inflammation and fibrosis, whereas NASH is accompanied by hepatic inflammation, necrosis, and fibrosis, and can progress to liver cirrhosis and hepatocellular carcinoma (HCC) [6]. NAFLD is closely associated with obesity, insulin resistance, type 2 diabetes mellitus (T2DM), hypertension, hyperlipidemia, and metabolic syndrome. The World Health Organization (WHO) defined overweight and obesity as body mass index (BMI) $\geq$25 and $\geq$30 kg/m$^2$, respectively. According to WHO global estimates, 39% and 13% of the adult population were overweight and obese, respectively, in 2016. The population of obese people is growing rapidly, with the worldwide prevalence of obesity having tripled between 1975 and 2016 [7]. The prevalence of NAFLD globally is estimated to be 25% [8]. It varies by geographic area, with the highest prevalence reported in the Middle East (32%) and South America (30%) and the lowest in Africa (13%). Ethnicity affects the vulnerability to NAFLD, which has a high prevalence in Hispanics and low prevalence in African Americans. A polymorphism in the patatin-like phospholipase domain containing 3 (*PNPLA3*) gene is one of the genetic factors that contribute to the geographic difference in NAFLD prevalence [4]. The *PNPLA3* gene encodes a triacylglycerol lipase that hydrolyzes triacylglycerol in adipocytes [9]. The frequency of the G (I148M) allele (rs738409) in this gene, which has been shown to be associated with increased liver fat content, was reported to be high in Hispanics and low in African Americans [4]. Along with the increase in the obese population, the population of NAFLD patients is also increasing [5, 10]. In China, the prevalence of NAFLD increased from 18.2% in 2000–2006 to 20.9% in 2010–2013 [11]. In a Japanese study analyzing almost 40,000 health checkup examinees, the prevalence of NAFLD diagnosed by abdominal ultrasonography rapidly increased from 12.6% in 1989 to 30.3% in 2000 [12].

NASH is an aggressive form of NAFLD. NASH, which can be diagnosed only by liver histology, can progress to liver cirrhosis, HCC, and liver failure. In the USA, NASH is already one of the most frequent etiologies for liver transplantation [3]. The prevalence of NASH is estimated to range from 6.7% to 29.9% in NAFLD patients and from 1.5% to 6.5% in the general population [2, 5]. Several studies have investigated the prognosis of NAFLD. The mortality of 435 patients with NAFLD diagnosed by imaging or liver biopsy was higher than the expected mortality of the general population (standardized mortality ratio 1.34, 95% confidence interval [CI] 1.003–1.76, P = 0.03) [13]. Patients with biopsy-proven NAFLD also had a higher mortality than the reference population (hazard ratio [HR] 1.29, 95% CI 1.04–1.59, P = 0.02) [14]. On the other hand, several studies (including meta-analysis) did not find an association between NAFLD and high mortality [15–17]. The most common cause of death in NAFLD patients is cardiovascular disease [14, 18]. In a cohort study including 229 patients with biopsy-proven

NAFLD with a mean follow-up period of 26.4 years, NAFLD was associated with an increased risk of death from cardiovascular disease compared with the reference population (HR 1.55, 95% CI 1.11–2.15, P = 0.01) [14]. However, a recent meta-analysis could not demonstrate the association of NAFLD with increased mortality from cardiovascular disease [17, 19]. Considering that controversies remain, more high-quality studies are required to clarify the outcome of NAFLD.

The existence of lean NAFLD is attracting much attention. Although lean NAFLD has not been clearly defined, BMI $<$25 kg/m$^2$ is generally applied. The prevalence of lean NAFLD was 7% in the USA, whereas a greater prevalence ($>$20%) was reported in Asian countries [4, 20]. Lean NAFLD is believed to have better prognosis than overweight/obese NAFLD [21]; however, a recent Japanese study [22] revealed that advanced fibrosis was not uncommon in lean NAFLD patients, especially in women. In a total of 762 patients with biopsy-proven NAFLD, advanced fibrosis (stage 3–4) was recognized in 31.0%, 41.6%, and 60.9% of lean ($<$25 kg/m$^2$), overweight (25–30 kg/m$^2$), and obese ($\geq$30 kg/m$^2$) men, respectively, and in 51.4%, 62.9%, and 33.7% of lean, overweight, and obese women, respectively. More studies are needed to elucidate the characteristics and natural history of lean NAFLD.

In this study, we analyzed the prognosis of 223 patients with biopsy-proven NAFLD with a median follow-up of 19.5 (range 0.5–41.0) years and 4248.3 person-years. We found that older age and the presence of T2DM as a comorbidity were significantly associated with all-cause mortality. In addition to the above factors, milder steatosis and advanced fibrosis were significant predictors of the occurrence of liver-related events. The prognosis was not significantly different between lean and overweight NAFLD patients.

## Methods

### Patients

In this retrospective cohort study, we reviewed consecutive patients who underwent liver biopsy from March 1975 to December 2012 at Tokai University Hospital, a referral hospital in the mid-western part of Kanagawa Prefecture with approximately 580,000 inhabitants. Most patients referred to our hospital due to liver dysfunction underwent liver biopsy to make a definite diagnosis. We enrolled those with a histological diagnosis of fatty liver. We excluded patients with other etiologies, such as (1) viral infection including hepatitis B virus (HBV) and hepatitis C virus (HCV); (2) autoimmune hepatitis (AIH), primary biliary cholangitis (PBC), and primary sclerosing cholangitis (PSC); (3) alcoholic liver diseases (ethanol consumption per week $\geq$210 g in men and $\geq$140 g in women); (4) drug-induced liver injury; (5) genetic liver diseases such as Wilson disease and hemochromatosis; (6) decompensated liver cirrhosis, liver failure, and HCC at baseline; (7) presence of uncontrolled extrahepatic malignancy or cardiovascular disease; and (8) follow-up period $<$6 months. This study was approved by the Institutional Review Board for Clinical Research at Tokai University (11R-222) and performed in accordance with the Declaration of Helsinki.

We collected information on age, sex, body weight, BMI, lifestyle, comorbidities, and laboratory data at the time of liver biopsy. We investigated the patients' current status if they continued to visit our hospital. We also conducted a questionnaire survey (S1 and S2 Files). The questionnaire was mailed to the patients and consisted of questions on their current status, lifestyle and comorbidities, and date and cause of death in mortality cases. From data collected from medical charts and questionnaires we analyzed the overall mortality, liver-related events (defined as the occurrence of HCC, decompensation, spontaneous bacterial peritonitis, variceal bleeding, and hepatic encephalopathy), and risk of comorbidities in this cohort. The last observation date was set to December 31, 2016. We accessed the database and medical records

from January 2017 to April 2019. The history of alcohol consumption was carefully investigated through medical charts and questionnaire responses. Alcohol consumption was also recorded at the first visit, at the time of liver biopsy, and returned visits, and patients consuming alcohol exceeding the criteria described above were excluded. Patients with a previous diagnosis of diabetes mellitus or with fasting glucose >126 mg/dL, hemoglobin A1c ≥6.5%, taking oral hypoglycemic agents, and those who revealed to be diabetic by questionnaire were defined as diabetics. Hypertension was defined as systolic blood pressure >140 mmHg and/or diastolic blood pressure ≥90 mmHg or requiring treatment or taking antihypertensive drugs, and those who revealed to be hypertension by questionnaire were defined as hypertension. Cardiovascular disease was defined as previously diagnosed coronary heart disease, cerebrovascular disease, peripheral vascular disease, heart failure, rheumatic heart disease, congenital heart disease, cardiomyopathies and those who revealed to be cardiovascular disease by questionnaire were defined as cardiovascular disease. Hypertriglyceridemia was defined as fasting triglyceride >150 mg/dL, and hypercholesterolemia was defined as total cholesterol >240 mg/dL, taking oral antihyperlipidemic drugs, and those who revealed to be hypertriglyceridemia or hypercholesterolemia by questionnaire were defined as hypertriglyceridemia or hypercholesterolemia. As part of this follow-up study all medical records were reviewed with special attention to information on alcohol consumption, and if viral hepatitis, especially hepatitis C virus, had been diagnosed during follow-up.The fibrosis-4 (FIB4) index was calculated using the following formula: age (years)×aspartate aminotransferase level (U/L)/platelet count ($10^9$/L) ×alanine aminotransferase (U/L) $^{1/2}$ [23].

## Liver histology

Liver biopsy was performed percutaneously or laparoscopically. Specimens stained with hematoxylin-eosin and azan stain were blindly re-evaluated by an experienced liver pathologist (K. M.). Liver histology findings were scored in accordance with the NASH Clinical Research Network scoring system developed by Kleiner et al [24]. NAFL and NASH were diagnosed according to the Practice Guidance from the American Association for the Study of Liver Diseases [2].

## Statistical analysis

Numerical and dichotomous baseline variables were compared using Student's t-test and chi-square test, respectively, between lean and overweight patients and between female and male patients. The time at risk was determined from the date of liver biopsy to the date of outcome or last follow-up. The incidence of HCC was analyzed using the person-years method. The all-cause mortality and occurrence of liver-related events were analyzed using Kaplan-Meier curves and log-rank test. The association of baseline variables with all-cause mortality and the occurrence of liver-related events were evaluated using a Cox proportional hazard model. The evaluated baseline variables included age, sex, BMI, alcohol intake, presence or absence of comorbidities (T2DM, hypertension), and pathological findings (steatosis, lobular inflammation, portal inflammation, ballooning, and fibrosis). Multivariate analysis was performed using the significant variables in univariate analysis. Multivariate analysis was performed using the significant variables in univariate analysis. The difference in the prevalence of T2DM and hypertension between NAFLD patients and the general population was evaluated using the standardized incidence ratio (SIR) and 95% CI based on the 2011 statistics database published by Kanagawa Prefecture [25], which was the place of residence of almost all patients. Analyses were performed using IBM SPSS Statistics version 25 software (IBM Corporation, Armonk, NY), and P values <0.05 were considered statistically significant.

### Ethics statement

This study was approved by the Institutional Review Board for Clinical Research at Tokai University (11R-222) and performed in accordance with the Declaration of Helsinki. A questionnaire was mailed to all subjects and written informed consent was obtained at the same time. Outpatients obtained written informed consent directly. For the deceased, their bereaved agreed on their behalf. Informed consent was obtained from all subjects for inclusion in the study. The research plan was posted on the website and bulletin board of the hospital.

## Results

A total of 6080 patients underwent liver biopsy from March 1975 to December 2012 (Fig 1). Of these, we reviewed 4095 patients who were followed up for 6 months or longer. We excluded 3872 patients with other etiologies such as HCV infection (n = 1321), HBV infection (n = 437), alcoholic liver disease (n = 469), AIH (n = 131), PBC (n = 104), PSC (n = 3), malignancy (n = 141), and undiagnosed chronic liver disease without steatosis (n = 1266). Finally, 223 patients with a histological diagnosis of NAFLD constituted the NAFLD cohort. Of these, 56 (25.1%) and 167 (74.9%) patients had NAFL and NASH, respectively. In NASH patients, the number of patients with fibrosis stage 0, 1, 2, 3, and 4 was 54 (32.3%), 40 (24.0%), 39 (23.4%), 25 (15.0%), and 9 (5.4%), respectively. We received a response to questionnaires from 111 patients (49.8%). Ninety patients (40.4%) were lost to follow-up.

The baseline demographic, clinical, and histological characteristics were compared between lean (BMI $<25$ kg/m$^2$) and overweight (BMI $\geq 25$ kg/m$^2$) patients (Table 1). The percentage of lean and overweight patients was 45.7% and 54.3%, respectively. The prevalence of T2DM was marginally greater in overweight patients (15.7% vs. 26.4%, P = 0.051). There were no significant differences in the prevalence of other comorbidities including dyslipidemia, hyperuricemia, hypertension, ischemic heart disease, and cerebrovascular disease. In liver histology, the incidence of NASH was 68.6% in lean patients, which was significantly smaller than that in

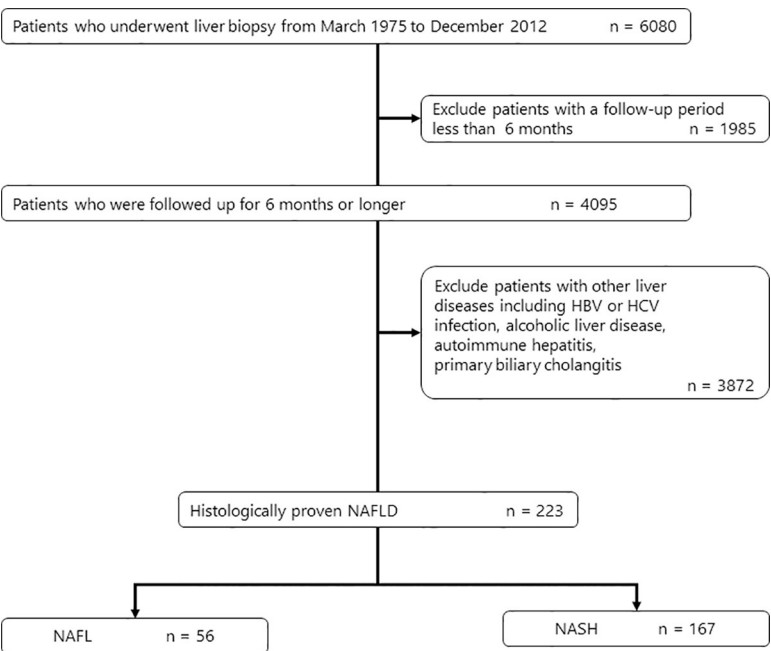

**Fig 1. Algorithm of patient selection.** NAFLD, nonalcoholic fatty liver disease; NAFL, nonalcoholic fatty liver; NASH, nonalcoholic steatohepatitis.

**Table 1. Baseline demographic, clinical and histological characteristics.**

| Variables | Lean [a] (n = 102) | | | | Overweight [b] (n = 121) | | | | |
|---|---|---|---|---|---|---|---|---|---|
| | Total | Female | Male | P value [c] | Total (n = 121) | Female | Male | P value [d] | P value [e] |
| | (n = 102) | (n = 33) | (n = 69) | | | (n = 42) | (n = 79) | | |
| Age (years) | 43.7 ± 14.1 | 49.8 ± 26.7 | 40.7 ± 11.8 | 0.002 | 42.9 ± 14.3 | 50.3 ± 12.9 | 39.0 ± 13.6 | < 0.001 | 0.7 |
| Weight (kg) | 59.6 ± 8.8 | 50.5 ± 6.5 | 63.9 ± 6.0 | < 0.001 | 76.5 ± 14.5 | 68.2 ± 8.9 | 81.0 ± 15.0 | < 0.001 | < 0.001 |
| BMI (kg/m$^2$) | 22.3 ± 2.1 | 21.5 ± 2.6 | 22.7 ± 1.7 | 0.005 | 30.0 ± 3.6 | 29.0 ± 2.9 | 28.9 ± 4.0 | 0.89 | < 0.001 |
| Alcohol intake [f] | 18 (17.6) | 1 (3.0) | 17 (24.6) | 0.007 | 18 (14.9) | 3 (7.1) | 15 (19.0) | 0.081 | 0.58 |
| Smoking | 41 (40.2) | 3 (9.1) | 38 (55.1) | < 0.001 | 39 (32.2) | 6 (14.3) | 33 (41.8) | 0.002 | 0.22 |
| Comorbidities | | | | | | | | | |
| Diabetes | 16 (15.7) | 6 (18.2) | 10 (14.5) | 0.63 | 32 (26.4) | 12 (28.6) | 20 (25.3) | 0.7 | 0.051 |
| Dyslipidemia | 8 (7.8) | 4 (12.1) | 4 (5.8) | 0.27 | 8 (6.6) | 1 (2.4) | 7 (8.9) | 0.26 | 0.72 |
| Hyperuricemia | 2 (2.0) | 0 (0.0) | 2 (2.9) | 1 | 5 (4.1) | 0 (0.0) | 5 (6.3) | 0.16 | 0.46 |
| Hypertension | 24 (23.5) | 10 (30.3) | 14 (20.3) | 0.27 | 32 (26.4) | 15 (35.7) | 17 (21.5) | 0.092 | 0.62 |
| Cardiovascular disease | 1 (1.0) | 0 (0.0) | 1 (1.4) | 1 | 5 (4.1) | 2 (4.8) | 3 (3.8) | 1 | 0.15 |
| Pathology | | | | | | | | | |
| NASH | 70 (68.6) | 26 (78.8) | 44 (63.8) | 0.13 | 97 (80.2) | 33 (78.6) | 64 (81.0) | 0.75 | 0.048 |
| Steatosis, grade ≥2 | 55 (53.9) | 16 (48.5) | 39 (56.5) | 0.45 | 87 (71.9) | 26 (61.9) | 61 (77.2) | 0.074 | 0.005 |
| Lobular inflammation, grade ≥2 | 15 (14.7) | 6 (18.2) | 9 (13.0) | 0.56 | 29 (25.0) | 12 (28.6) | 17 (21.5) | 0.39 | 0.083 |
| Portal inflammation, grade ≥2 | 15 (14.7) | 8 (24.2) | 7 (10.1) | 0.076 | 28 (23.1) | 14 (33.3) | 14 (17.7) | 0.053 | 0.11 |
| Ballooning, grade ≥1 | 75 (73.5) | 24 (72.7) | 51 (73.9) | 0.9 | 112 (92.6) | 39 (92.9) | 73 (92.4) | 1 | < 0.001 |
| Fibrosis, stage ≥3 | 8 (7.8) | 6 (18.2) | 2 (2.9) | 0.013 | 28 (23.1) | 17 (40.5) | 11 (13.9) | 0.001 | 0.002 |

Data are presented as mean ± standard deviation or number (%) of patients.

a. Lean, BMI < 25 kg/m$^2$

b. Overweight, BMI ≥ 25 kg/m$^2$

c. Female vs male in lean patients.

d. Female vs male in overweight patients.

e. Lean patients vs overweight patients.

f. Alcohol intake < 210 g/week for men and < 140 g/week for women.

overweight patients (80.2%, P = 0.048). Grade ≥2 steatosis, stage ≥3 fibrosis, and presence of ballooning was less frequently observed in lean patients than in overweight patients: grade ≥2 steatosis, 53.9% vs. 71.9% (P = 0.005); stage ≥3 fibrosis: 7.8% vs. 23.1% (P = 0.002); and presence of ballooning; 73.5% vs. 92.6% (P<0.001). Details of histological findings including steatosis, lobular inflammation, portal inflammation, ballooning, and fibrosis, are shown in S1 Table. With respect to age, female patients were significantly older by approximately 10 years than male patients in both the lean and overweight groups. The proportions of drinkers and smokers in female patients were smaller than those in male patients. Notably, the incidence of advanced fibrosis was significantly greater in female patients than in male patients irrespective of whether they were lean or overweight (lean patients: 18.2% vs. 2.9%, P = 0.013; overweight patients: 40.5% vs. 13.9%, P = 0.001). In laboratory data, white blood cell count was lower in lean patients (mean ± SD: 5990 ± 1745 /μL vs. 6743 ± 2014 /μL, P = 0.009; S2 Table). The serum transaminase levels, albumin, platelet count, and FIB4 index did not significantly differ between lean and overweight patients. When compared between female and male patients, white blood cell count and serum albumin level were lower and the FIB4 index was higher in female patients than in male patients.

We calculated the SIR of T2DM and hypertension in comparison with the general population (Table 2). The risk of T2DM was 11.7 (95% CI 8.70–15.6) times higher in NAFLD patients

**Table 2. Standardized incidence ratio (SIR) of diabetes and hypertension in NAFLD patients compared with the general population.**

| Comorbidities | Total | Lean [a] | Overweight [b] |
|---|---|---|---|
| | (n = 223) | (n = 102) | (n = 121) |
| Diabetes | 11.7 (8.70–15.6) | 8.16 (4.83–13.6) | 14.8 (10.3–21.2) |
| Hypertension | 7.99 (6.09–10.5) | 6.88 (4.51–10.4) | 9.09 (6.32–13.0) |

Each value represents SIR (95% confidence interval) in the NAFLD patients when SIR in the general population is adjusted to 1.

a. Lean, BMI $<25$ kg/m$^2$

b. Overweight, BMI $\geq 25$ kg/m$^2$.

than in the general population. The risk of hypertension was also higher in NAFLD patients (7.99 [95% CI 6.09–10.5]) than in the general population. Although the risk of T2DM and hypertension was higher in overweight patients than in lean patients, the difference was not significant. The median duration of follow-up for this cohort was 19.5 (range, 0.5–41.0) years, with a total of 4248.3 person-years of follow-up (PYF). A total of 23 patients died, most of whom had NASH (22 patients, 95.7%). The causes of death were extrahepatic malignancy (n = 5, 21.7%); cardiovascular disease (n = 5, 21.7%); liver-related causes (n = 3, 13.0%) including HCC (n = 1, 4.3%); other causes including respiratory disease, renal disease, and duodenal ulcer (n = 5, 21.7%); and unknown (n = 5, 21.7%). The number of all-cause deaths per 1000 PYF was 0.92 (95% CI 0.16–5.2) in NAFL patients, whereas it was 7.0 (95% CI 4.6–10.6) per 1000 PYF in NASH patients (Table 3). Although not significant, overweight, T2DM, and advanced fibrosis caused greater all-cause mortality calculated per 1000 PYF than their counterparts (overweight: 3.6 [95% CI 1.8–7.1] vs. 7.5 [95% CI 4.5–12.3], T2DM: 4.0 [95% CI 2.4–6.7] vs. 10.5 [95% CI 5.5–19.9], and advanced fibrosis: 4.6 [95% CI 2.9–7.2] vs. 13.8 [95% CI: 6.3–30.1]).

We analyzed the variables related to all-cause mortality. NASH patients showed significantly poorer survival than NAFL patients (P = 0.041, Fig 2A). We also analyzed the association of baseline variables with survival. In univariate analysis, older age, female sex, BMI $\geq 25$ kg/m$^2$ (Fig 2B), presence of T2DM, and fibrosis stage $\geq 3$ were associated with poor survival (Table 4). Advanced fibrosis (stage 3–4) was associated with poorer survival than stage 0 (HR 5.0, 95% CI 1.69–14.9, P = 0.004). In multivariate analysis, however, T2DM was the only

**Table 3. All-cause mortality and liver-related events per 1000 person-year of follow-up.**

| Variables | | Patients number (n) | All-cause mortality | | | Liver-related events | | |
|---|---|---|---|---|---|---|---|---|
| | | | PYF [a] | Death (n) | Per 1000 PYF (95% CI) [b] | PYF | Events (n) | Per 1000 PYF (95% CI) |
| BMI (kg/m$^2$) | $< 25$ | 102 | 2234.6 | 8 | 3.6 (1.8–7.1) | 2153.3 | 5 | 2.3 (0.99–5.4) |
| | $\geq 25$ | 121 | 2013.7 | 15 | 7.5 (4.5–12.3) | 1969.8 | 9 | 4.6 (2.4–8.7) |
| Diabetes | no | 175 | 3390.1 | 14 | 4.0 (2.4–6.7) | 3321.9 | 7 | 2.1 (1.0–4.4) |
| | yes | 48 | 858.2 | 9 | 10.5 (5.5–19.9) | 801.1 | 7 | 8.7 (4.2–18.0) |
| NAFLD | NAFL | 56 | 1092.1 | 1 | 0.92 (0.16–5.2) | 1062.3 | 1 | 0.94 (0.17–5.3) |
| | NASH | 167 | 3156.2 | 22 | 7.0 (4.6–10.6) | 3060.7 | 13 | 4.3 (2.5–7.3) |
| Steatosis grade | $< 2$ | 81 | 1564.8 | 10 | 6.4 (3.5–11.8) | 1471.5 | 10 | 6.8 (3.7–12.5) |
| | $\geq 2$ | 142 | 2683.5 | 13 | 4.8 (2.8–8.3) | 2651.5 | 4 | 1.5 (0.60–3.9) |
| Fibrosis stage | $\leq 2$ | 187 | 3813.0 | 17 | 4.6 (2.9–7.2) | 3721.6 | 6 | 1.6 (0.74–3.5) |
| | $\geq 3$ | 36 | 435.3 | 6 | 13.8 (6.3–30.1) | 401.4 | 8 | 19.9 (10.1–39.3) |

a PYF, person-year of follow-up

b CI, confidence interval

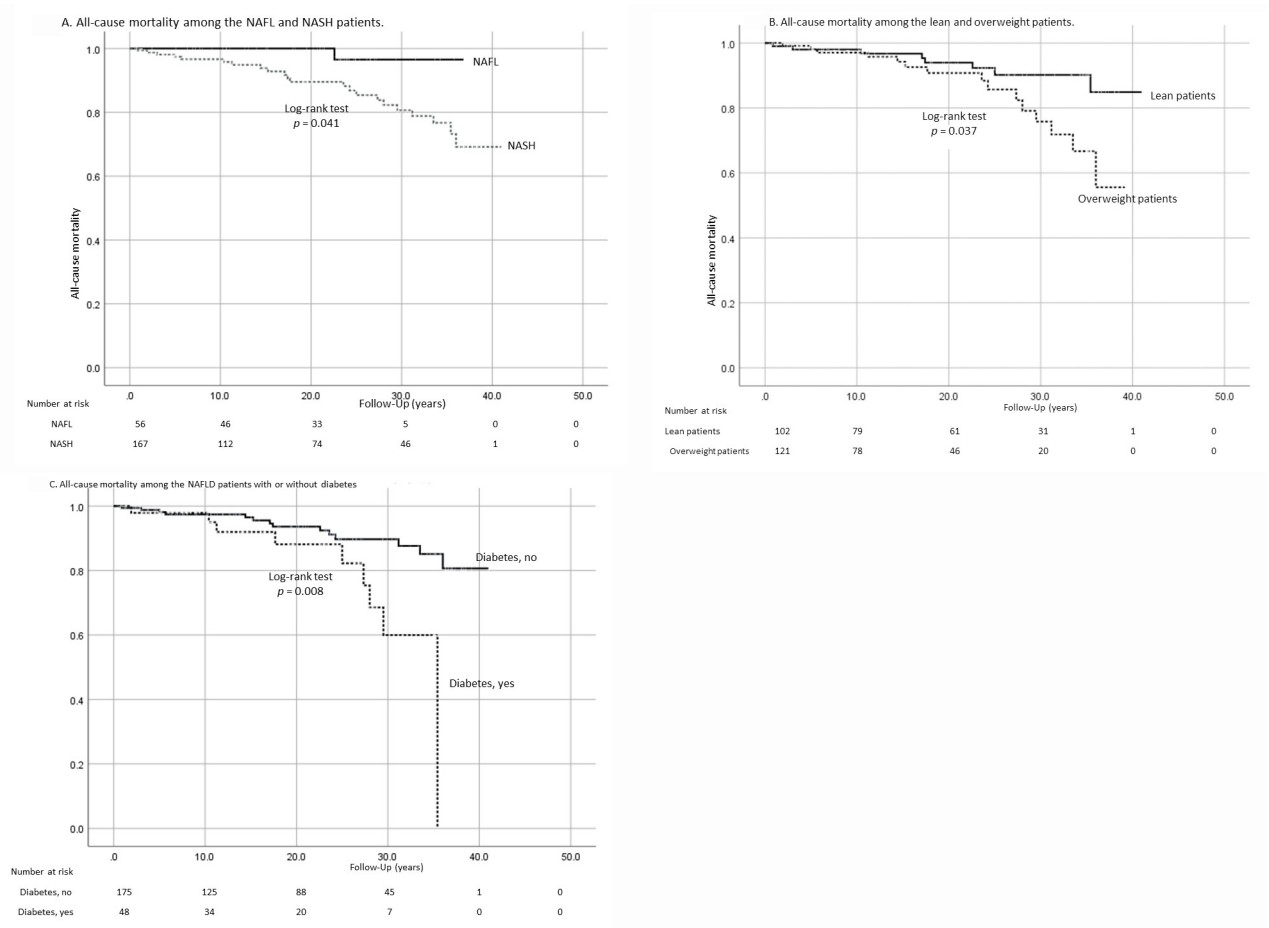

**Fig 2. All-cause mortality in patients with nonalcoholic fatty liver disease (NAFLD).** (A) All-cause mortality in NAFL and NASH patients. (B) All-cause mortality in lean and overweight NAFLD patients. (C) All-cause mortality in NAFLD patients with and without type 2 diabetes mellitus.

significant variable associated with all-cause mortality (HR 2.87, 95% CI 1.12–7.04, P = 0.021; Table 4, Fig 2C). A total of 14 liver-related events were observed. Patients with advanced fibrosis more frequently experienced liver-related events than those without advanced fibrosis (1.6 [95% CI 0.74–3.5] vs. 19.9 [95% CI 10.1–39.3] per 1000 PYF, Table 3). Six patients developed HCC, all of whom had NASH. One of these patients died of HCC. The incidence of HCC was analyzed using the person-year method because the number of patients who developed HCC was small. HCC occurred at 0.77 (95% CI 0.26–2.25) and 9.88 (95% CI 3.36–29.1) per 1000 PYF in patients with fibrosis stage 0–2 and in those with fibrosis stage 3–4, respectively. Therefore, the incidence of HCC was significantly higher in patients with advanced fibrosis. Further, we analyzed the association of baseline variables with the occurrence of liver-related events (Table 5). The occurrence of liver-related events was not different between lean and overweight patients even in univariate analysis. Multivariate analysis revealed that older age, presence of T2DM (Fig 3A), milder hepatic steatosis (grade 0–1, Fig 3B), and advanced liver fibrosis (stage 3–4, Fig 3C) were significantly associated with the occurrence of liver-related events. BMI $\geq$22 kg/m$^2$ was also analyzed as one of the predictors of all-cause mortality, occurrence of liver-related events, and occurrence of HCC, but it was shown that, like BMI $\geq$25 kg/m$^2$, it was not a significant predictor (S3–S6 Tables, S1 and S2 Figs).

**Table 4. Factors associated with all-cause mortality among the NAFLD patients.**

| Variables [a] | | Univariate | | | Multivariate [b] | | |
|---|---|---|---|---|---|---|---|
| | | HR [c] | 95% CI [d] | P value | HR | 95% CI | P value |
| Age (years) | | 1.09 | 1.05–1.14 | < 0.001 | 1.09 | 1.05–1.14 | < 0.001 |
| Gender, male | | 0.34 | 0.15–0.78 | 0.011 | 0.72 | 0.29–1.8 | 0.49 |
| BMI $\geq$ 25 kg/m$^2$ | | 2.44 | 1.03–5.79 | 0.044 | 2.35 | 0.93–5.92 | 0.071 |
| Comorbidities | | | | | | | |
| Diabetes, yes | | 3.02 | 1.28–7.13 | 0.012 | 2.87 | 1.12–7.04 | 0.021 |
| Pathological findings | | | | | | | |
| Fibrosis stage | 0 | | reference | | | reference | |
| | 1–2 | 1.34 | 0.53–3.5 | 0.54 | 1.39 | 0.52–3.68 | 0.51 |
| | 3–4 | 5 | 1.69–14.9 | 0.004 | 2.39 | 0.71–8.0 | 0.16 |

a. Listed are variables that are significantly associated with all-cause mortality by univariate analysis.

b. Multivariate analysis is performed using variables significantly associated with all-cause mortality by univariate analysis.

c. HR, hazard ratio

d. CI, confidence interval

## Discussion

Through a long-term follow-up study of biopsy-proven NAFLD over 19 years, we demonstrated the following findings: (1) the prevalence of T2DM and hypertension was higher in NAFLD patients than in the general population; (2) the most common causes of death were extrahepatic malignancy and cardiovascular disease followed by liver-related mortality; (3) older age and T2DM were associated with all-cause mortality; (4) older age, T2DM, milder steatosis, and advanced fibrosis were associated with the occurrence of liver-related events; and (5) the prognosis was not significantly different between lean and overweight patients.

Of 223 patients who underwent liver biopsy, 25% and 75% patients were diagnosed with NAFL and NASH, respectively. Considering that NASH represents 7–30% of NAFLD [8], the percentage of 75% seems high. This discrepancy may be explained by the presence of a selection bias. Patients requiring liver biopsy were assumed to have relatively advanced liver disease. In fact, a meta-analysis showed that the prevalence of NASH among NAFLD patients

**Table 5. Factors associated with the occurrence of liver-related events among the NAFLD patients.**

| Variables [a] | | Univariate | | | Multivariate [b] | | |
|---|---|---|---|---|---|---|---|
| | | HR [c] | 95% CI [d] | P value | HR | 95% CI | P value |
| Age (years) | | 1.11 | 1.06–1.17 | < 0.001 | 1.12 | 1.04–1.19 | 0.001 |
| Gender, male | | 0.12 | 0.033–0.43 | 0.001 | 0.33 | 0.08–1.26 | 0.10 |
| Comorbidities | | | | | | | |
| Diabetes, yes | | 3.96 | 1.39–11.3 | 0.010 | 6.08 | 1.79–20.7 | 0.004 |
| Pathological findings | | | | | | | |
| Steatosis, grade $\geq$ 2 | | 0.22 | 0.068–0.70 | 0.010 | 0.23 | 0.07–0.77 | 0.018 |
| Fibrosis stage | 0 | | reference | | | reference | |
| | 1–2 | 1.14 | 0.23–5.64 | 0.87 | 1.53 | 0.29–8.22 | 0.62 |
| | 3–4 | 10.8 | 2.81–41.4 | 0.001 | 5.87 | 1.41–24.4 | 0.015 |

a. Listed are variables that are significantly associated with liver-related events by univariate analysis.

b. Multivariate analysis is performed using variables significantly associated with the occurrence of liver-related events by univariate analysis.

c. HR, hazard ratio

d. CI, confidence interval

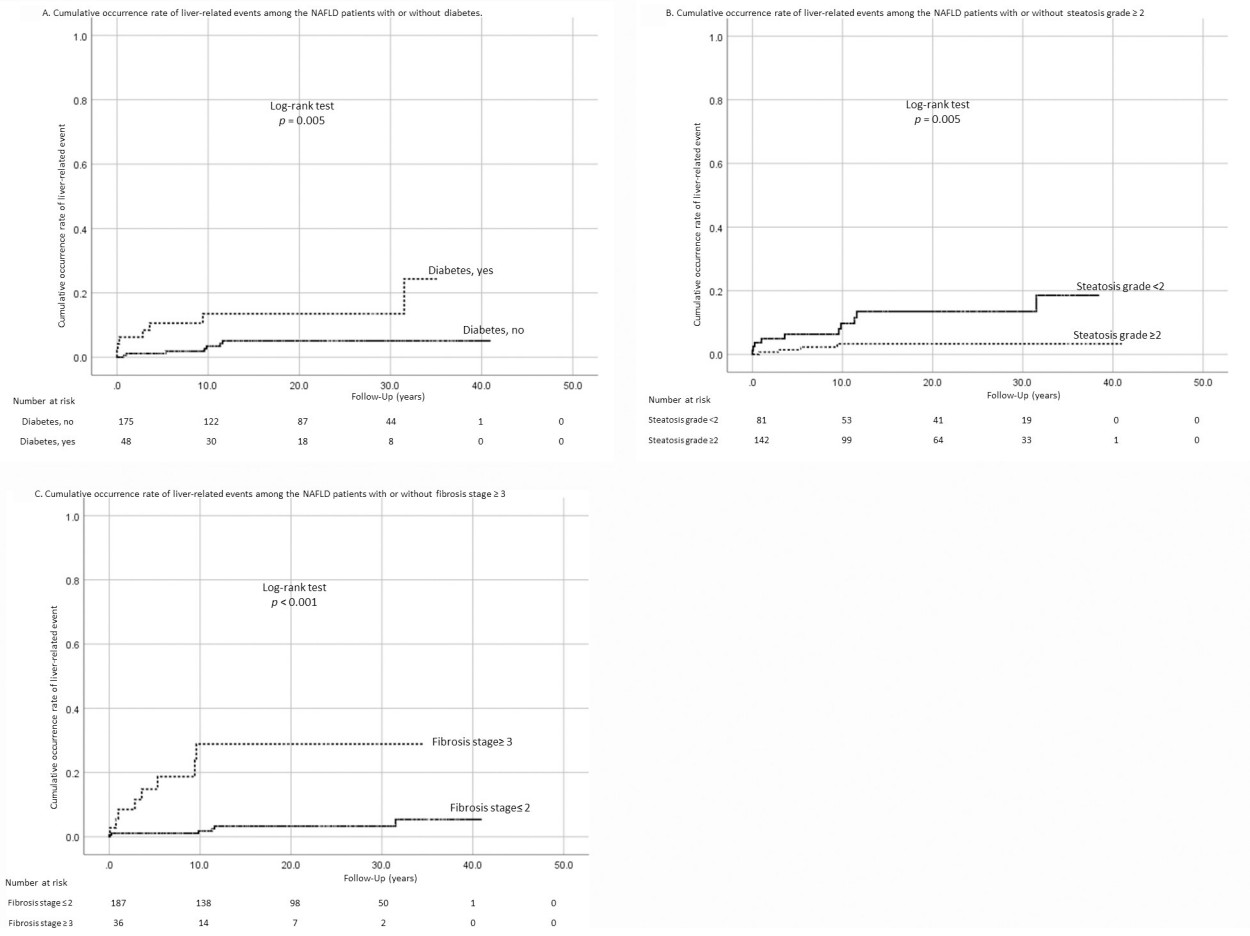

**Fig 3. Cumulative occurrence rate of liver-related events.** (A) Cumulative occurrence rate of liver-related events in patients with nonalcoholic fatty liver disease (NAFLD) with and without type 2 diabetes mellitus. (B) Cumulative occurrence rate of liver-related events in NALFD patients with grade <2 steatosis and those with grade ≥2 steatosis. (C) Cumulative occurrence rate of liver-related events in NALFD patients with stage ≤2 fibrosis and in those with stage ≥3 fibrosis.

with an indication for biopsy were 63.45% (95% CI 47.68–76.79) in Asia, 69.25% (95% CI 55.93–79.98) in Europe, and 60.64% (95% CI 49.56–70.72) in North America [8]. The proportion of overweight individuals was marginally greater in the NASH group than in the NAFL group (58.1% vs. 42.9%, P = 0.048; S1 Table), although the difference in BMI was not statistically significant. The aforementioned meta-analysis [8] revealed that the obesity prevalence estimates were 51.3% (95% CI 41.4–61.2) among NAFLD patients and 81.8% (95% CI 55.2–94.3) among NASH patients. The obesity prevalence among the NASH patients in our cohort might be lower than that reported in the meta-analysis. This difference may be caused by the differences in patient characteristics, such as ethnicity.

As comorbidities of NAFLD in this cohort, the prevalence of T2DM and hypertension was 11.7 (95% CI 8.70–15.6) and 7.99 (95% CI 6.09–10.5) times greater, respectively, than in the general population. This observation agrees with the concept of NAFLD as a hepatic phenotype of metabolic syndrome. Although the prevalence of T2DM and hypertension in overweight patients was greater than that in lean patients, the difference was not significant. Therefore, this suggests that NAFLD patients have a higher risk of T2DM and hypertension even if they are not obese.

A total of 23 patients in our cohort died, with most of them (22 patients) having NASH. The most common causes of death were extrahepatic malignancy (21.7%) and cardiovascular disease (21.7%) followed by liver-related mortality (13.0%). These results are in agreement with a number of previous studies demonstrating that cardiovascular disease [14, 16, 18, 21, 26–29], extrahepatic malignancy [13, 14, 30, 31], and liver-related mortality [15, 32, 33] were the leading causes of death in NAFLD patients. Notably, only 1 (1.8%) of 56 patients with NAFL died during the 19 years follow-up, whereas 22 (13.2%) of 167 patients with NASH died. Therefore, NAFL might be a benign disease [15]. In multivariate analysis, older age (HR 1.09, 95% CI 1.05–1.14) and the presence of T2DM (HR 2.87, 95% CI 1.12–7.04) were the factors associated with all-cause mortality. Several studies associated T2DM with all-cause mortality with an HR ranging from 1.6 to 2.7 [13, 14, 26, 27, 29]. T2DM is well known to increase the incidence of neoplasms [34], presumably through the continuous increase in serum insulin and insulin-like growth factors levels [35]. T2DM is also associated with the incidence of cardiovascular disease [36] Adams et al. [14] conducted a community-based cohort study comparing 116 diabetic patients with NAFLD (most were diagnosed by imaging) with 221 diabetic patients without NAFLD. According to their study, NAFLD was significantly associated with overall mortality (HR 2.3, 95% CI 1.1–4.2, P = 0.03) but was not associated with mortality either from malignancy (HR 2.3, 95% CI 0.9–5.9, P = 0.09) or cardiovascular disease (HR 0.9, 95% CI 0.3–2.4, P = 0.81). A recent meta-analysis [37] including 14 studies reached the same conclusion. In brief, NAFLD increased the all-cause mortality (HR 1.34, 95% CI 1.17–1.54), whereas it did not increase the mortality from cardiovascular disease (HR 1.13, 95% CI 0.92–1.38) and malignancy (HR 1.05, 95% CI 0.89–1.25). Therefore, not NAFLD itself but NAFLD coexisting T2DM may largely contribute to the increase in mortality from cardiovascular disease and malignancy.

In our study, older age, T2DM, milder steatosis, and advanced fibrosis were associated with the occurrence of liver-related events. Many studies associated advanced fibrosis with all-cause mortality and the risk of liver-related events [14, 18, 27, 28]. A recent meta-analysis [38] demonstrated that both all-cause and liver-related mortality risks increased along with the progression of fibrosis. In our study, fibrosis was extracted as a significant predictive variable for liver-related events but not for all-cause mortality. The reason of this discrepancy is unclear, but the difference in ethnicity might have influenced the outcomes. In the study of Ekstedt et al. [14], advanced fibrosis was significantly associated with all-cause mortality, but this association was no longer significant when analysis was performed after excluding T2DM patients. These results suggest a strong association between T2DM and mortality in NAFLD as others [13] and we demonstrated (Table 6).

Very few studies have investigated the relationship between hepatic steatosis and mortality. According to Angulo et al. [27], the extent of steatosis was not associated with either all-cause mortality or liver-related events. We found a weak relation between hepatic fibrosis and steatosis: steatosis decreased along with the progression of fibrosis (Pearson's correlation coefficient, r = 0.163, P = 0.013, S7 Table). Therefore, reduced steatosis with moderate fibrosis may be a factor leading to "burnt-out" NASH, which is considered as late stage NASH accompanying loss of hepatic fat [13, 39]. Further investigations are necessary to elucidate the impact of steatosis on the prognosis of NAFLD.

Patients with T2DM had a six times greater risk of liver-related events and all-cause mortality than those without T2DM. T2DM not only increases the risk of HCC occurrence but also accelerates the progression of fibrosis, although the mechanism by which T2DM induces fibrosis remains unclear. Adams et al. [13] analyzed the histological changes in 103 NAFLD patients who underwent serial liver biopsies and demonstrated that T2DM, high BMI, and mild fibrosis at baseline were factors associated with fibrosis progression. Tada et al. [40] analyzed the

**Table 6. Cohort studies on histologically diagnosed NAFLD.**

| Study | Location Time period | NAFLD/ NASH n (%) | Men n (%) | Age (years) mean±SD | BMI (kg/ m² mean ±SD | Comorbidities n (%) | Follow-up (years) median (range) | Fibrosis stages 0/1/2/3/4 (%) | Variables associated with prognosis all-cause mortality HR (95%CI) | liver-related mortality HR (95%CI) | Reference |
|---|---|---|---|---|---|---|---|---|---|---|---|
| Younossi et al. | USA | NAFL 78 (37) | 79 (38) | NAFLD 48.1 ± 15.3 | NAFL 34.9 ± 10.8 | diabetes 43 (21) | 12.2 (IQR[a] 4.9–15.5) | 23/28/ 14/22/13 | not mentioned | advanced fibrosis: HR 5.7 (1.5–21.5) | 50 |
| retrospective | not listed | NASH 131 (63) | | NASH 49.1 ± 14.4 | NASH 37.4 ± 10.3 | hyperlipidemia 47 (23) | | | | | |
| Ekstedt et al. | Sweden | NAFLD 229 | 149 (65) | 48.8 ± 12.8 | 28.3 ± 3.7 | diabetes 31 (14) | mean ± SD | F0-2/3-4: | advanced fibrosis (F3-4): HR 3.3 (2.3–4.8) | advanced fibrosis: HR 10.8 (1.4–83.9) | 14 |
| prospective | 1980-1993 | | | | | hypertension 130 (57) | 26.4 ± 5.6 | 88.8/11.2 | | | |
| | | | | | | hyperlipidemia 78 (34) | | | | | |
| Angulo et al. | Multinational | NAFL 335 (54) | 232 (38) | median (range) 49 (38–60) | median (range) 30.7 (26.4–36.5) | diabetes 232 (38) | 12.6 (0.3–35.1) | 52/23/ 14/9/3 | Fibrosis stage 0: reference | Fibrosis stage 0: reference | 27 |
| retrospective | 1975-2005 | NASH 284 (46) | | | | hypertension 190 (31) | | | stage 1: HR 1.8 (1.2–2.8) | stage 1: HR 2.4 (0.6–8.9) | |
| | | | | | | | | | stage 2: HR 1.9 (1.2–3.0) | stage 2: HR 7.5 (2.3–24.9) | |
| | | | | | | | | | stage 3: HR 1.9 (1.2–3.1) | stage 3: HR 13.8 (4.4–43.7) | |
| | | | | | | | | | stage 4: HR 6.4 (3.4–12.0) | stage 4: HR 47.5 (11.9–189) | |
| Adams et al. | USA | NAFLD 435 biopsy (n = 65) NASH 49 (75) | 231 (49) | 49 ± 15 | 33.5 ± 6.5 | diabetes 299 (26) | 7.6 (0.1–23.5) | biopsy (n = 65) | age(per decade): HR 2.2(1.7–2.7) | not mentioned | 13 |
| prospective | 1980–2000 | | | | | hypertension 155 (36) | | 36/18/ 15/18/13 | IFG[b]/ diabetes: HR 2.6 (1.3–5.2) | | |
| | | | | | | | | | cirrhosis: HR 3.1 (1.2–7.8) | | |
| Hirose et al. | Japan | NAFL 56 (25) | 148 (66) | 43.3 ± 14.2 | 25.9 ± 4.5 | diabetes 48 (22) | 19.5 (0.5–41.0) | 43/23/ 19/12/4 | age (per years): HR 1.1 (1.05–1.1) | Liver-related event | this paper |
| retrospective | 1975–2016 | NASH 167 (75) | | | | hypertension 56 (25) | | | diabetes: HR 2.9 (1.1–7.0) | age (per year): HR 1.1 (1.04–1.2) | |
| | | | | | | | | | | diabetes: HR 6.1 (1.8–21) | |
| | | | | | | | | | | milder steatosis: HR 4.4 (1.3–15.2) | |
| | | | | | | | | | | advanced fibrosis: HR 5.9 (1.4–24) | |

a IQR, interquartile range

b IFG; Impaired fasting glucose

predictive factors for advanced fibrosis by following up 1562 NAFLD patients diagnosed by imaging with less severe fibrosis for a median of 7.5 years. A total of 186 patients progressed to advanced fibrosis, which was determined by a FIB-4 index of >2.67. T2DM was independently associated with fibrosis progression (HR 1.88, 95% CI 1.40–25.2, P<0.001) as well as age ≥50 years and serum albumin concentration <4.2 g/dL. Therefore, controlling T2DM is crucial to improve the prognosis of NAFLD.

In our study, HCC occurrence was more frequently observed in patients with advanced fibrosis (stage 3–4, 9.88 [95% CI 3.36–29.1]/1000 PYF) than in those with mild fibrosis (stage 0–2, 0.77 [95% CI 0.26–2.25]/1000 PYF). Kanwal et al. [41] conducted a large retrospective cohort study comparing the risk of HCC between almost 0.3 million NAFLD patients diagnosed according to a predefined algorithm and the same number of matched controls. The HCC incidence in NAFLD patients was significantly higher (0.21/1000 PYF) than that in controls (0.02/1000 PYF), with the highest risk among those with NAFLD cirrhosis (10.6/1000 PYF). Our results are in concordance with this previous study. Advanced fibrosis significantly contributes to the incidence of HCC in NAFLD patients, as already shown in viral hepatitis [42].

Lean NAFLD is now drawing considerable attention. Leung et al. [21] prospectively followed up 72 lean (BMI <25 kg/m$^2$) and 235 overweight (BMI ≥25 kg/m$^2$) biopsy-proven NAFLD patients. Lean patients had a lower grade of steatosis and lower stage of fibrosis than overweight patients. The event-free survival was also better in lean patients (P = 0.019). We could not find a significant difference in the clinical outcome between lean and overweight patients. Advanced fibrosis was more frequently observed in lean NAFLD women than in lean NAFLD men (18.2% vs. 2.9%, P = 0.013). In accordance with this observation, the serum albumin level was lower and the FIB4 index was higher in women than in men. Notably, in our cohort, women were older than men by approximately 10 years. Although aging could affect the progression of fibrosis, women might be vulnerable to fibrosis progression even if they are not obese [22]. The prognosis of lean NAFLD patients is not necessarily better than that of overweight NAFLD patients, especially in women. Further studies are required to elucidate the outcome in lean NAFLD patients.

This study has several limitations. First, this was a retrospective cohort study performed at a single hospital. The cohort consisted of a relatively small number of Japanese patients. The outcome of NAFLD would be different according to ethnicity and geographic areas, which likely harbor different lifestyles, habits, and genetic variations. Hence, our results may not represent the characteristics and outcome of general NAFLD. Second, this study was a referral hospital-based study, which is likely to yield a selection bias: the patient composition could deviate to those with advanced liver disease. We did not use matched controls to evaluate the prognosis of NAFLD patients. Third, as this study had no controls, we could not calculate relative risk for the occurrence of T2DM and hypertension. Instead, we calculated an SIR with the 2011 Kanagawa Prefecture database as a reference. The prevalence of hypertension had been unchanged [43], while the T2DM prevalence had been increasing [44, 45] in the past 50 years in Japan. Our study had a long-term entry period (1975–2012). Therefore, the actual SIR of T2DM might be further greater than the calculated SIR. Forth, we had difficulty in investigating medical charts as old as 40 years. Several important laboratory data, including platelet counts and albumin levels, were missing and could not be used to analyze their contribution to the outcome. Fifth, only half of the patients sent back the questionnaires and approximately 40% of the patients were lost to follow-up. This would occur mostly due to moving and death. Considerable lack of data could have influenced the outcomes, however, loss of the patients might be unavoidable in a long-term cohort study. Finally, we could not analyze the impact of T2DM severity and treatment on the outcome of NALFD patients. Glucose-lowering therapies

may differentially influence the cancer risk. Metformin, which decreases systemic insulin levels, is reported to reduce the cancer risk [46] and cancer-related mortality [47], although recent papers [48, 49] are challenging the results of previous studies. A far greater number of patients is needed to elucidate the effect of T2DM severity and treatment on the prognosis.

The strength of our study was the long-term follow-up period, with median and maximum follow-up of 19.5 and 41.0 years, respectively. NAFLD studies with a follow-up period of >10 years are still few [14–16, 26–29, 50, 51] and were not reported in Asia. As NAFLD is a heterogeneous disease and its prevalence and prognosis would be different according to ethnicity and geographical area, our study would contribute to understanding the characteristics of NAFLD especially in Asian patients. Another strength is that our patients were diagnosed by liver biopsy and histology performed by an experienced liver pathologist. Despite the advances in imaging modalities, the diagnosis of NASH still relies on liver histology. Attention should be paid to the modality used to make a diagnosis when evaluating NAFLD studies. In conclusion, through a long-term follow-up study of 223 biopsy-proven NAFLD patients, we revealed that the presence of T2DM as a comorbidity and older age were significantly associated with all-cause mortality. In addition to the above-mentioned factors, milder steatosis and advanced fibrosis were significant predictors of the occurrence of liver-related events. The prognosis of lean NAFLD patients is not necessarily better than that of overweight NAFLD patients.

## Supporting information

**S1 File.**
(DOCX)

**S2 File.**
(DOCX)

**S1 Questionnaire.**
(DOCX)

**S2 Questionnaire.**
(DOCX)

**S1 Answer.**
(DOCX)

**S2 Answer.**
(DOCX)

**S1 Fig. All-cause mortality among the BMI ≥22 and <22.**
(TIF)

**S2 Fig. Cumulative occurrence rate of liver-related events among the BMI ≥22 and <22.**
(TIF)

**S1 Table. Baseline histological characteristics.** Date are presented as the number (%) of patients.
(XLSX)

**S2 Table. Baseline laboratory data.** Each value represents mean ± standard deviation. The number of patients who have available data is shown in parenthesis.
(XLSX)

**S3 Table. Factors associated with all-cause mortality among the NAFLD patients, including BMI ≥ 22.**
(XLSX)

**S4 Table. Factors associated with the occurrence of liver-related events among the NAFLD patients, including BMI ≥ 22.**
(XLSX)

**S5 Table. All-cause mortality and Liver-related events per 1000 person-year of follow-up, BMI 22 instead of BMI 25.**
(XLSX)

**S6 Table. Incidence of HCC per 1000 person-year of follow-up.**
(XLSX)

**S7 Table. Correlation of clinical factors and pathological factors.**
(XLSX)

## Acknowledgments

We thank members of Pathology Department and Support Center for Medical Research and Education, Tokai University for technical assistance.

## Author Contributions

**Conceptualization:** Shunji Hirose, Tatehiro Kagawa.

**Data curation:** Kota Tsuruya, Kazuya Anzai, Yoshitaka Arase, Koichi Shiraishi, Michiko Suzuki, Satsuki Ieda.

**Investigation:** Koshi Matsumoto, Masayuki Tatemichi.

**Supervision:** Tatehiro Kagawa.

**Writing – original draft:** Shunji Hirose.

**Writing – review & editing:** Shunji Hirose.

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
