## [Decision Letter · Decision Letter 0]

3 Aug 2020

PONE-D-20-21370

Nineteen-year prognosis in Japanese patients with biopsy-proven nonalcoholic fatty liver disease: Lean versus overweight patients

PLOS ONE

Dear Dr. Hirose,

Thank you for submitting your manuscript to PLOS ONE. After careful consideration, we feel that it has merit but does not fully meet PLOS ONE’s publication criteria as it currently stands. Therefore, we invite you to submit a revised version of the manuscript that addresses the points raised during the review process.

While the study is of a potential interest, the reviewers identified several important issues such as a large amount of people, who were lost to follow-up. These need to be eliminated or at least properly acknowledged. In addition, multiple large studies have been carried out on this subject. The authors should compare their findings with these studies (a table might be reasonable way to do this) and discuss the discrepancies.   

We look forward to receiving your revised manuscript.

Kind regards,

Pavel Strnad

Academic Editor

PLOS ONE

Journal Requirements:

2. Please provide additional details regarding participant consent. In the ethics statement in the Methods and online submission information, please ensure that you have specified (1) whether consent was informed for people completing the questionnaire by mail, and (2) what type of consent you obtained for outpatients (for instance, written or verbal, and if verbal, how it was documented and witnessed).

3. Please include the date(s) on which you accessed the databases or records to obtain the data used in your study.

4. Please include additional information regarding the survey or questionnaire used in the study and ensure that you have provided sufficient details that others could replicate the analyses. For instance, if you developed a questionnaire as part of this study and it is not under a copyright more restrictive than CC-BY, please include a copy, in both the original language and English, as Supporting Information.

5. Our internal editors have looked over your manuscript and determined that it is within the scope of our Liver Diseases Call for Papers. This collection of papers is headed by a team of Guest Editors for PLOS ONE. Additional information can be found on our announcement page: https://collections.plos.org/s/liver-diseases. If you would like your manuscript to be considered for this collection, please let us know in your cover letter and we will ensure that your paper is treated as if you were responding to this call. If you would prefer to remove your manuscript from collection consideration, please specify this in the cover letter.

6. Thank you for stating the following financial disclosure:

"No"

7. We note that you have included the phrase “data not shown” in your manuscript. Unfortunately, this does not meet our data sharing requirements. PLOS does not permit references to inaccessible data. We require that authors provide all relevant data within the paper, Supporting Information files, or in an acceptable, public repository. Please add a citation to support this phrase or upload the data that corresponds with these findings to a stable repository (such as Figshare or Dryad) and provide and URLs, DOIs, or accession numbers that may be used to access these data. Or, if the data are not a core part of the research being presented in your study, we ask that you remove the phrase that refers to these data.

8. Your ethics statement must appear in the Methods section of your manuscript. If your ethics statement is written in any section besides the Methods, please move it to the Methods section and delete it from any other section. Please also ensure that your ethics statement is included in your manuscript, as the ethics section of your online submission will not be published alongside your manuscript.

9. Please upload copies of Figure 1, 2, 3, to which you refer in your text. If the figures are no longer to be included as part of the submission please remove all reference to it within the text.

Reviewers' comments:

Reviewer's Responses to Questions

**Comments to the Author**

1. Is the manuscript technically sound, and do the data support the conclusions?

Reviewer #1: Yes

Reviewer #2: Partly

2. Has the statistical analysis been performed appropriately and rigorously? 

Reviewer #1: Yes

Reviewer #2: I Don't Know

3. Have the authors made all data underlying the findings in their manuscript fully available?

Reviewer #1: No

Reviewer #2: No

4. Is the manuscript presented in an intelligible fashion and written in standard English?

Reviewer #1: Yes

Reviewer #2: Yes

5. Review Comments to the Author

Reviewer #1: Overall, this is an interesting and well written paper. Most NALD studies are suffering from short follow ups. In this retrospective monocenter study Hirose et al. examined a long-term cohort of biopsy proven NAFLD in 223 patients over 19 years. They concluded that NASH, older age, and T2DM were significantly associated with all-cause mortality. Factors significantly associated with liver-related events were older age, T2DM, milder hepatic steatosis, and advanced liver fibrosis. As an add on they found that the prognosis was not significantly different between lean and overweight NAFLD patients.

Major issues:

From 223 only 111 (less than 50%) answered and send back the questionnaires. This fact has to be discussed in more detail with regard to medication, drug abuse, sports.

The percentage of lean and overweight patients was 45.7% and 54.3%, respectively. This seems to be special because other studies describe more obese patients. Is it a phenomenon observed in Japan or Asia only?

The authors should discuss, why NASH cirrhosis was not present in their patients.

Minor issues:

The following sentences of the abstract should be better placed in the result section than under methods: “The risk of type 2 diabetes mellitus (T2DM) and hypertension was 11.7 (95% confidence interval [CI] 8.70-15.6) and 7.99 (95% CI 6.09-10.5) times higher, respectively, in NAFLD patients than in the general population. Twenty-three patients died, 22 of whom had NASH.”

Limitations of the study are mentioned. The terminus of “burnt-out” NASH with regard to steatosis with moderate fibrosis should be explained in more detail.

The questionnaire should be added - at least as supplemental material.

Reviewer #2: The submitted manuscript by Hirose et al. describes the 19 year long term prognosis of obese and non-obese patients with NAFLD in a cohort of 223 patients. The worldwide increasing prevalence of NAFLD related mortality implicates the relevance of research in this field. The authors conclude, that T2DM is highly prevalent in NAFLD patients and was correlated to mortality and liver related events. Furthermore they state, that lean patients prognosis appears to be not necessarily better than in obese patients. Several issues flaw the reviewers enthusiam about this manuscript. There are numerous multinational NAFLD cohort studies of significantly larger scale than it is the case in this manuscript, examining the long term effects of NAFLD in different ethnicites. The overall novelty of the authors findings is very limited. The majority of messages in this manuscript have been published previously. Furthermore, as the authors have already stated, there is a relevant selection bias comprising only one quarter of the patients to have NAFLD, while three quarters of patients suffer from NASH, making the cohort not represenative of real life data.

Minor comments:

The authors should have stated what type of questionnaires were utilized for alcohol and health-related issues.

Histological findings appear to be based on HE staining proven fibrosis. Relevant information about NAFLD activity score, ie. extent of steatoisi, lobular inflammation, balloning, etc. is not provided.

Despite the overall small number of patients enrolled in this study the rather high rate of patients lost to FUP of 40% further decreasing the statistical power of this study.

6. PLOS authors have the option to publish the peer review history of their article (what does this mean?). If published, this will include your full peer review and any attached files.

Reviewer #1: No

Reviewer #2: No

---

## [Author Response · Author response to Decision Letter 0]

3 Oct 2020

First of all, thank you very much for reviewing our paper. We have revised our manuscript according to the comments of the editor and reviewers. We described point-to-point response.

To the editor’s comments;

While the study is of a potential interest, the reviewers identified several important issues such as a large amount of people, who were lost to follow-up. These need to be eliminated or at least properly acknowledged. In addition, multiple large studies have been carried out on this subject. The authors should compare their findings with these studies (a table might be reasonable way to do this) and discuss the discrepancies.

Thank you for your valuable suggestion. According to the suggestion, we have made Table 6 listing previous major papers and added the sentence discussing about the difference between other papers and ours (line 332-338).

We have formatted according to the journal’s style. 

2. Please provide additional details regarding participant consent. In the ethics statement in the Methods and online submission information, please ensure that you have specified (1) whether consent was informed for people completing the questionnaire by mail, and (2) what type of consent you obtained for outpatients (for instance, written or verbal, and if verbal, how it was documented and witnessed).

We described clearly about the consent (line 177-178).

3. Please include the date(s) on which you accessed the databases or records to obtain the data used in your study.

 We described the accession dates (line 131).

4. Please include additional information regarding the survey or questionnaire used in the study and ensure that you have provided sufficient details that others could replicate the analyses. For instance, if you developed a questionnaire as part of this study and it is not under a copyright more restrictive than CC-BY, please include a copy, in both the original language and English, as Supporting Information.

 We added the questionnaire in Japanese and in English as Supporting information.

7. We note that you have included the phrase “data not shown” in your manuscript. Unfortunately, this does not meet our data sharing requirements. PLOS does not permit references to inaccessible data. We require that authors provide all relevant data within the paper, Supporting Information files, or in an acceptable, public repository. Please add a citation to support this phrase or upload the data that corresponds with these findings to a stable repository (such as Figshare or Dryad) and provide and URLs, DOIs, or accession numbers that may be used to access these data. Or, if the data are not a core part of the research being presented in your study, we ask that you remove the phrase that refers to these data.

 We have added all data as S3-S6 tables and S1, S2 figures.

―――――――――――――――――――――――――――

Reviewer #1: Overall, this is an interesting and well written paper. Most NALD studies are suffering from short follow ups. In this retrospective monocenter study Hirose et al. examined a long-term cohort of biopsy proven NAFLD in 223 patients over 19 years. They concluded that NASH, older age, and T2DM were significantly associated with all-cause mortality. Factors significantly associated with liver-related events were older age, T2DM, milder hepatic steatosis, and advanced liver fibrosis. As an add on they found that the prognosis was not significantly different between lean and overweight NAFLD patients.

Major issues:

From 223 only 111 (less than 50%) answered and send back the questionnaires. This fact has to be discussed in more detail with regard to medication, drug abuse, sports.

Thank you for your valuable suggestion. This is a limitation of our study. We discussed about this limitation (line 396-399) as follows, “Only half of the patients sent back the questionnaires and approximately 40% of the patients were lost to follow-up. This would occur mostly due to moving and death. Considerable lack of data could have influenced the outcomes, however, loss of the patients might be unavoidable in a long-term cohort study.”

The percentage of lean and overweight patients was 45.7% and 54.3%, respectively. This seems to be special because other studies describe more obese patients. Is it a phenomenon observed in Japan or Asia only?

Thank you for your valuable suggestion. As we described in line 95-96, the prevalence of lean NAFLD is higher in Asian countries than Western countries. 

The authors should discuss, why NASH cirrhosis was not present in their patients.

Thank you for your suggestion. There are 9 patients with F4 fibrosis. Therefore, the percentage of cirrhotic patients is 4% in our total cohort as we wrote in line 188-189.

Minor issues:

The following sentences of the abstract should be better placed in the result section than under methods: “The risk of type 2 diabetes mellitus (T2DM) and hypertension was 11.7 (95% confidence interval [CI] 8.70-15.6) and 7.99 (95% CI 6.09-10.5) times higher, respectively, in NAFLD patients than in the general population. Twenty-three patients died, 22 of whom had NASH.”

Thank you for your suggestion. According to your suggestion, we have moved the above sentence to Result section. 

Limitations of the study are mentioned. The terminus of “burnt-out” NASH with regard to steatosis with moderate fibrosis should be explained in more detail.

Thank you for your suggestion. We have added the explanation of “burnt-out” NASH (line 347-348).

The questionnaire should be added - at least as supplemental material.

Thank you for your suggestion. We have added the questionnaire in Japanese and in English as Supporting information.

Reviewer #2: 

The submitted manuscript by Hirose et al. describes the 19 year long term prognosis of obese and non-obese patients with NAFLD in a cohort of 223 patients. The worldwide increasing prevalence of NAFLD related mortality implicates the relevance of research in this field. The authors conclude, that T2DM is highly prevalent in NAFLD patients and was correlated to mortality and liver related events. Furthermore they state, that lean patients prognosis appears to be not necessarily better than in obese patients. Several issues flaw the reviewers enthusiam about this manuscript. There are numerous multinational NAFLD cohort studies of significantly larger scale than it is the case in this manuscript, examining the long term effects of NAFLD in different ethnicites. The overall novelty of the authors findings is very limited. The majority of messages in this manuscript have been published previously. Furthermore, as the authors have already stated, there is a relevant selection bias comprising only one quarter of the patients to have NAFLD, while three quarters of patients suffer from NASH, making the cohort not represenative of real life data.

Thank you for your valuable comments. We understand that there are many studies analyzing prognosis of NAFLD. However, long-term follow-up studies over 10 years on biopsy-proven NAFLD are still few as we mentioned in Discussion (line 405-407). Therefore, we believe that our study would provide useful information to journals’ readers.

Minor comments:

The authors should have stated what type of questionnaires were utilized for alcohol and health-related issues.

 Thank you for your suggestion. We have added the questionnaire in Japanese and in English as Supporting information.

Histological findings appear to be based on HE staining proven fibrosis. Relevant information about NAFLD activity score, ie. extent of steatosis, lobular inflammation, ballooning, etc. is not provided. 

Thank you for your suggestion. We have added the detailed data of histology as S1 table. 

Despite the overall small number of patients enrolled in this study the rather high rate of patients lost to FUP of 40% further decreasing the statistical power of this study.

Thank you for your valuable suggestion. This is a limitation of our study. We discussed about this limitation (line 396-399) as follows, “Only half of the patients sent back the questionnaires and approximately 40% of the patients were lost to follow-up. This would occur mostly due to moving and death. Considerable lack of data could have influenced the outcomes, however, loss of the patients might be unavoidable in a long-term cohort study.”

---

## [Decision Letter · Decision Letter 1]

21 Oct 2020

Nineteen-year prognosis in Japanese patients with biopsy-proven nonalcoholic fatty liver disease: Lean versus overweight patients

PONE-D-20-21370R1

Dear Dr. Hirose,

We’re pleased to inform you that your manuscript has been judged scientifically suitable for publication and will be formally accepted for publication once it meets all outstanding technical requirements.

Kind regards,

Pavel Strnad

Academic Editor

PLOS ONE

Additional Editor Comments (optional):

Reviewers' comments:

Reviewer's Responses to Questions

**Comments to the Author**

1. If the authors have adequately addressed your comments raised in a previous round of review and you feel that this manuscript is now acceptable for publication, you may indicate that here to bypass the “Comments to the Author” section, enter your conflict of interest statement in the “Confidential to Editor” section, and submit your "Accept" recommendation.

Reviewer #1: All comments have been addressed

Reviewer #2: All comments have been addressed

2. Is the manuscript technically sound, and do the data support the conclusions?

Reviewer #1: Yes

Reviewer #2: Yes

3. Has the statistical analysis been performed appropriately and rigorously? 

Reviewer #1: Yes

Reviewer #2: I Don't Know

4. Have the authors made all data underlying the findings in their manuscript fully available?

Reviewer #1: Yes

Reviewer #2: No

5. Is the manuscript presented in an intelligible fashion and written in standard English?

Reviewer #1: Yes

Reviewer #2: Yes

6. Review Comments to the Author

Reviewer #1: All concers have been removed by the authors response. The mansucript is now ready to publish.

The authors submitted important additonal material in the supplement.

Reviewer #2: The paper significantly improved in direct comparison to the previous version. Therefore it can be accepted for publication in its current form

7. PLOS authors have the option to publish the peer review history of their article (what does this mean?). If published, this will include your full peer review and any attached files.

Reviewer #1: No

Reviewer #2: No

---

## [Editor Report · Acceptance letter]

29 Oct 2020

PONE-D-20-21370R1 

Nineteen-year prognosis in Japanese patients with biopsy-proven nonalcoholic fatty liver disease: Lean versus overweight patients 

Dear Dr. Hirose:

I'm pleased to inform you that your manuscript has been deemed suitable for publication in PLOS ONE. Congratulations! Your manuscript is now with our production department. 

Kind regards, 

on behalf of

Dr. Pavel Strnad 

Academic Editor

PLOS ONE